# Climate and Bedrock Collectively Influence the Diversity Pattern of Plant Communities in Qiniangshan Mountain

**DOI:** 10.3390/plants13243567

**Published:** 2024-12-20

**Authors:** Xujie Li, Wanyi Zhao, Xianling Sun, Xuejiao Zhang, Wenbo Liao, Qiang Fan

**Affiliations:** 1State Key Laboratory of Biocontrol and Guangdong Provincial Key Laboratory of Plant Resources, School of Life Sciences, Sun Yat-sen University, Guangzhou 510275, China; lixj267@mail3.sysu.edu.cn (X.L.); zhaowy25@mail.sysu.edu.cn (W.Z.); 2Shenzhen Dapeng Peninsula National Geopark, Shenzhen 518116, China; m13530288757@163.com (X.S.); zxj160501330@163.com (X.Z.)

**Keywords:** plant community, diversity pattern, species richness, phylogenetic structure

## Abstract

Climate and geological diversity have been proven to make an important contribution to biodiversity. Volcanic ecosystems often have a long geological history and diverse bedrock, thus shaping a variety of habitats. Understanding the relative importance and role of the contemporary climate and geological bedrock environment in volcanic biodiversity still needs further exploration. To address this knowledge gap, we investigated the patterns of plant diversity and phylogenetic structure at the community level in Qiniangshan Mountain, while also exploring the relationship between biodiversity and regional environmental factors (e.g., climate and bedrock types). In the Qiniangshan Mountain plant communities, species richness is higher at mid-to-high elevations. Montane communities exhibit higher species richness compared to coastal communities. There are significant differences in species richness among plant communities on different bedrock, with the highest species richness found on pyroclastic lava. Bedrock, along with climate factors related to energy and precipitation, collectively influence the patterns of species richness in plant communities. The Net Relatedness Index (NRI) of plant communities is influenced by climate factors and aspects, while the Nearest Taxon Index (NTI) is affected by both bedrock and climate factors. The Phylogenetic Diversity Index (PDI) is primarily related to climate factors. Climate and bedrock collectively influence the patterns of species richness and phylogenetic structure within Qiniangshan Mountain’s plant communities. These findings highlight the profound impact of both climate and bedrock on montane vegetation and community biodiversity.

## 1. Introduction

The biogeographic patterns of mountain biodiversity are influenced by climatic and geological factors across a range of temporal and spatial scales [1,2,3]. Climate is acknowledged as a critical environmental factor affecting biodiversity along latitudinal and elevational gradients [4,5]. Geological diversity contributes to habitat heterogeneity and influences local biodiversity by generating ecological niches [6,7,8].

Climate is one of the most significant environmental factors affecting biodiversity patterns. In the study of plant diversity patterns at regional and larger spatial scales, plant diversity is more affected by the climate in space and time than by other factors [5]. However, on the local spatial scale, the impact of rapid climate change on the diversity of local plant communities is not unique. For example, the study of mountain plant communities in temperate regions of northern Europe shows that climate warming will lead to a longer growing season and higher productivity of species in mountain plant communities, and the species richness of the communities will also increase [9], which may be related to the diffusion of low-altitude species to high altitudes [10]. However, for the mountainous areas in the Mediterranean region of Europe, climate warming will lead to a water shortage, which will significantly reduce the species richness of plant communities [11]. Therefore, the response of plant community species diversity to climate factors in different regions may be different. There is a close interdependence between climate variables (e.g., temperature and rainfall) and plants [4], and it has been proven that regions with warm and humid climates have higher species diversity than regions with cold or dry climates [12,13]. Climate change can cause changes in the plant community’s structure. For example, climate warming affects the community composition and distribution of the entire Himalayan region, including alpine meadows and grasslands, wetlands, peatlands and forests [14]. Geological and geomorphic features can also significantly influence the region’s biodiversity. The erosion of mountain bedrock has created complex topographies, offering new ecological niches for a variety of life forms and positioning mountains as hotspots of terrestrial biodiversity [15,16,17]. Among the geological factors, the type and abundance of bedrock are intimately connected to species richness and composition at the local level. Bedrock influences plant diversity through its effects on soil chemistry and physical properties, promoting the emergence of endemic plant species within the region [18,19]. Soils originating from the weathering of various bedrock types and local hygrothermal conditions affect plant species distribution and community composition differentiation. Limestone and serpentine are prominent examples of such influences [20,21,22]. Bedrock influences the mineral nutrient content (e.g., P, Fe and Mg) and water-holding capacity of the regolith by regulating the physical and chemical properties of the soil, which in turn influence plant growth and the composition of vegetation communities [23,24]. Correspondingly, the heavy metals (e.g., Hg, Pb and Cd) provided by bedrock also inhibit plant growth in communities [25]. In addition, a higher bedrock fracture density can promote vegetation community formation by providing footholds for the root systems of plants and reservoirs for water storage [26,27].

Investigating community assembly mechanisms is a pivotal direction for understanding plant responses to current climatic and environmental changes [28,29]. Traditional biodiversity studies within communities have tended to concentrate on species diversity, often neglecting the evolutionary historical differences and phylogenetic diversity among species. With the advancement of “community phylogenetics” in recent years, the molecular dating of phylogenetic trees and subsequent biogeographic and diversification statistical analyses have become routine in biodiversity and evolutionary history research [30,31,32]. Phylogenetic diversity connects the evolutionary history of species with the conservation of functional diversity, reflecting the adaptive capacity of species to environmental changes. By integrating species richness with phylogenetic diversity, studies of plant community biodiversity can more effectively elucidate the patterns of plant diversity and formulate appropriate conservation strategies [33,34].

The niche theory of community assembly posits that species within a community occupy different niches, with the assembly process primarily driven by environmental filtering and competitive exclusion [35,36,37]. Based on the niche conservatism theory (e.g., rapid in situ speciation, dispersal limitation), species that are phylogenetically closer tend to have higher ecological similarity. When the phylogenetic structure of a community is clustered, it indicates that environmental filtering is selected for species with similar niches, thus habitat filtering dominates the community assembly process. Conversely, when the phylogenetic structure is overdispersed, it suggests that competitive exclusion prevents ecologically similar species from coexisting, leading to a community where species are more distantly related, and competitive exclusion drives the assembly process [38]. Furthermore, when niches converge (e.g., convergent evolution, random immigration), a clustered phylogenetic structure may indicate that dominant species occupy the primary niches, with competitive exclusion driving the assembly process. On the other hand, a divergent phylogenetic structure under environmental pressure suggests that distantly related species are undergoing convergent evolution, with habitat filtering dominating the assembly process [39]. The neutral theory of community assembly posits that interactions among species at the individual level are equivalent, and community assembly is a stochastic process of ecological drift [40]. In studies of phylogenetic community structure, the null model assumes species randomization from the community species pool. Similar to the neutral theory, when phylogenetic distances align with null model results, the community’s phylogenetic structure tends to be random, indicating that the assembly process follows the neutral theory. Therefore, by examining the phylogenetic structure of a community, one can validate the niche theory and neutral theory of community assembly, revealing the dominant factors in the assembly process and elucidating the mechanisms of species coexistence within the community.

The complex bedrock landform in Qiniangshan Mountain is derived from the violent volcanic eruption during the Yanshan Movement in the past. In order to explore the influence of environmental factors on the species diversity of the local volcanic ecosystem, we examined the community species richness and phylogenetic diversity patterns of 75 plant communities in Qiniangshan Mountain. Our goals were as follows: (i) to reveal the diversity pattern of plant communities; (ii) to estimate the effect of environmental factors on the species richness and phylogenetic structure of plant communities.

## 2. Materials and Methods

### 2.1. Study Area

Qiniangshan Mountain, a geological remnant of the Yanshan orogeny, is located in the southeastern part of the Dapeng Peninsula in Shenzhen, Guangdong Province, China. Its geographical coordinates range from 22°29′31″ N to 22°33′21″ N and from 114°31′14″ E to 114°37′22″ E, covering an area of 46.073 km^2^, with an elevation range from sea level to 869.7 m a.s.l. The bedrock in Qiniangshan Mountain mainly consists of volcanic rocks, including rhyolite, pyroclastic rock and pyroclastic lava, accompanied by granite and quartz sandstone (Figure 1). In addition, Qiniangshan Mountain is characterized by a typical South Asian subtropical maritime monsoon climate, with an average annual temperature ranging from 21 to 22 °C and annual precipitation of approximately 2280 mm (Meteorological Bureau of Shenzhen Municipality, http://weather.sz.gov.cn/ accessed on 18 December 2024). Under the influence of strong sea breezes and sunlight, the southeastern slope of Qiniangshan Mountain is predominantly covered by evergreen deciduous broad-leaved forests, dwarf forests and shrubs. In contrast, the northwestern slope is characterized by evergreen broad-leaved forests and ravine monsoon rainforests. Qiniangshan Mountain exhibits a remarkable diversity of vegetation, rendering it an ideal site for investigating the intricate interplay between flora and the bedrock environment. The maps of China and Qiniangshan Mountain used in this study were adapted from the DataV.GeoAtlas (http://datav.aliyun.com/portal/school/atlas/area_selector accessed on 8 October 2024), with visualization in ArcGIS 10.8 (http://www.esri.com/ accessed on 9 October 2024).

### 2.2. Community Data Acquisition and Standardization

The investigated communities were situated within the undisturbed natural vegetation zone of Qiniangshan Mountain. The field survey spanned from July 2020 to August 2021 to study the difference in vegetation communities on different bedrock types. We strategically placed community quadrats to comprehensively account for geographical and environmental factors (e.g., elevation, slope, aspect and bedrock). The survey method of the field community quadrat was adopted based on previous studies [41]. The sampled quadrat area of plant communities was 400 m^2^–1200 m^2^. For each community, we recorded its geographical data (e.g., latitude, longitude and elevation) (Extended Data Appendix A) and all species occurrences’ information. A total of 594 species belonging to 119 families and 322 genera were recorded in 75 plant communities (Extended Data Appendix A).

The ‘lcvplants’ package in R 4.2.3 (https://www.r-project.org/ accessed on 20 October 2024) was utilized to standardize the scientific names of plants in the community quadrats to the Leipzig Catalogue of Vascular Plants (LCVP), and taxonomic ranks below the species level were consolidated to the species level [42].

### 2.3. Environmental Data

(1)Climate data

The 19 bioclimatic variables, BIO1-BIO19, were retrieved from the CHELSA climate dataset (version 1.2, available at http://chelsa-climate.org/ accessed on 20 October 2024) at 30 arc-second resolution, which describes temperature, precipitation and fluctuations in temperature and precipitation at various time scales [43]. Although these precise climate data were rough for each community, they still had a certain correlation with the species richness of the community for the whole Qiniangshan Mountain region (Extended Data Appendix A). The extraction of climatic data was conducted using the ‘raster’ package in R version 4.2.3 [44]. Because some of the climatic variables (e.g., mean annual temperature and mean temperature of the warmest quarter) were highly intercorrelated, we excluded variables with a Pearson correlation coefficient greater than 0.95 to reduce collinearity. Seven climatic variables were finally included in our analysis of plant community quadrats (Extended Data Appendix A).

(2)Bedrock data

The bedrock in the study area was categorized into five types based on classification standards from *Volcanic Petrology* [45] and the geological survey findings of Qiniangshan Mountain [46]. These are three volcanic rocks: rhyolite (lava), pyroclastic rock and pyroclastic lava; one subvolcanic rock: granite; and one sedimentary rock: quartz sandstone (Figure 1). Rhyolite, pyroclastic rock, pyroclastic lava and granite stem from Mesozoic volcanic eruptions and contribute to Qiniangshan Mountain’s complex and varied volcanic topography. The quartz sandstone, dating back to the Devonian period, exclusively occurs in the Luzui area situated southeast of Qiniangshan Mountain (Figure 1). The largest bedrock in Qiniangshan Mountain is rhyolite, followed by pyroclastic rock, both present in the 0–800 m elevation range. Pyroclastic lava is concentrically distributed in the 100–600 m elevation range. Conversely, granite and quartz sandstone are primarily found below 100 m elevation.

(3)Topographic data

The topographic dataset includes three variables: slope, aspect and elevation. These data were obtained from field measurements by GPSMAP 62sc.

### 2.4. Phylogenetic Diversity and Phylogenetic Structure

(1)Phylogenetic tree construction

Utilizing standardized species information from community surveys, we constructed a phylogenetic tree with the ‘V.PhyloMaker2’ package in R 4.2.3 [47]. The backbone of the tree, GBOTB.extended.LCVP.tre, was based on the LCVP nomenclature standardization system and included gene sequence data with GenBank [42,48,49]. Among the 594 species occurring in plant communities, 458 species were included in “GBOTB.extended.LCVP.tre”, respectively. The other species absent from “GBOTB.extended.LCVP.tre” were matched to their respective genera based on the Phylomatic method by ‘V.PhyloMaker2’ [50]. Branch lengths for these species were then estimated using the BLADJ function in ‘V.PhyloMaker2’ [47,50]. We used build. nodes. 1 to extract genus and family information and generated our final mega-phylogeny using Scenario 3 in V.PhyloMaker2 following Jin and Qian (2022) [47]. Finally, we generated phylogenetic trees for 594 species in 75 plant communities in Qiniangshan Mountain (Extended Data Appendix A).

(2)Phylogenetic diversity and phylogenetic structure

We selected the most frequently used phylogenetic diversity (PD) metric to quantify community phylogenetic diversity [51]. PD is the cumulative sum of all phylogenetic branch lengths that connect the species within a community [51]. PD is significantly positively correlated with species richness (*p* < 0.001) (Extended Data Appendix A). To mitigate the effect of species richness on PD, we used the Phylogenetic Diversity Index (PDI), which is the standardized effect size of PD’s null model. The PDI is calculated under the null model with PD as [PDI = (PD_observed_ − PD_randomized_)/(sdPD_randomized_)] by ‘picante’ in R version 4.2.3 [52]. Here, PD_observed_ represents the observed PD, PD_randomized_ represents the expected PD obtained from 999 random combinations and sdPD_randomized_ is the standard deviation of the expected PD.

The inference of the community phylogenetic structure was based on the Net Relatedness Index (NRI), and the Nearest Taxon Index (NTI). The NRI measures the phylogenetic relatedness among species within a community [33,53,54]. Based on a null model, it is the standardized effect size of the mean phylogenetic distance (MPD) of all species pairs [30]. The NRI reflects the deeper phylogenetic structure. Based on a null model, the NTI is the standardized effect size of the mean nearest taxon distance (MNTD) for each species, indicating the shallower phylogenetic relationships. The calculated NRI and NTI were under the same null model, with the NRI as [NRI = − (MPD_observed_ − MPD_randomized_)/(sdMPD_randomized_)] and NTI as [NTI = − (MNTD_observed_ − MNTD_randomized_)/(sdMNTD_randomized_)] via the R package ‘PhyloMeasures’ [55].

### 2.5. Statistical Analysis

Data normalization: The species richness data of plant community quadrats were transformed using the Hellinger method [56]. Quantitative variables, including elevation, slope and climatic factors were processed using the range normalization formula, denoted as (x_i_ − min)/(max − min), effectively scaling the data to a range between 0 and 1 [57].

GLM regression analysis: Because the bedrock effects can be confounded with local climate and habitat conditions, we extended the generalized linear models (GLM) to incorporate all environmental data, including bedrock type, elevation, slope, aspect and climatic factors, as predictors in regression analyses of SR, NRI, NTI and PDI. Model selection during regression was progressively based on the Akaike information criterion (AIC) values, favoring models with lower AIC values. These regressions were conducted using the ‘MASS’ package and the ‘glm’ function in R version 4.2.3.

Spatial autocorrelation: Considering the spatial scale constraints on species dispersal, the species composition of communities exhibited spatial autocorrelation. The homogenizing effect of dispersal between two communities intensifies with proximity [58,59,60]. Thus, this study utilized Moran’s I value to measure the residual spatial autocorrelation, akin to Pearson’s correlation coefficient, which typically ranges from 1 to −1. An expected Moran’s I value close to 0 suggests an absence of spatial autocorrelation. The spatial autoregressive model (SAR) regression, performed with the ‘spdep’ package in R version 4.2.3, accounts for spatial autocorrelation interference among plant community quadrats, thereby evaluating the impact of various environmental factors [61]. For detailed model output results, refer to Extended Data Appendix A.

## 3. Results

### 3.1. Diversity Pattern of Plant Communities

In Qiniangshan Mountain, plant communities with higher species richness (SR) are primarily distributed in the mid-to-high-elevation range of approximately 400–600 m a.s.l. (Figure 2a). Conversely, plant communities with lower species richness are primarily located in low-elevation areas below 300 m a.s.l. Furthermore, there is a notable difference in the species richness of plant communities on the north and south slopes of Qiniangshan Mountain. Montane communities on the northwestern slope connected with the land exhibit higher species richness, while coastal communities on the southeastern slope near the ocean have lower species richness. The species richness of plant communities in Qiniangshan Mountain exhibits a “monotonically increasing” pattern along the elevation gradient (Figure 2b). The species richness of plant communities at mid-to-high elevations is significantly higher than that of low-elevation communities (R^2^ = 0.18, *p* < 0.001). There is no significant linear relationship between species richness and elevation for plant communities on different bedrock types (Figure 2c, *p*-values > 0.05). Overall, montane communities have higher species richness compared to coastal communities. However, there is no significant linear relationship between species richness and elevation in mountain communities (R^2^ = 0.10, *p* = 0.062), whereas coastal communities exhibit a significantly “monotonically increasing” pattern of species richness along the elevation gradient (R^2^ = 0.19, *p* = 0.006).

The species diversity of the community is, to some extent, influenced by the underlying bedrock. In Qiniangshan Mountain, the species richness of plant communities distributed on different bedrocks shows significant differences (Figure 3a). Plant communities on pyroclastic lava exhibit significantly higher species richness compared to those on granite (*p* = 0.013), pyroclastic rock (*p* = 0.011) and quartz sandstone (*p* = 0.0098). Additionally, plant communities on rhyolite have significantly higher species richness than those on pyroclastic rock (*p* = 0.0057) and quartz sandstone (*p* = 0.00065). An analysis of the differences in the community phylogenetic structure indicates that the NRI of plant communities on pyroclastic rock is significantly lower than those on granite (*p* = 0.043), quartz sandstone (*p* = 0.048) and rhyolite (*p* = 0.0049) (Figure 3b). The deep phylogenetic structure of plant communities on pyroclastic rock tends to be overdispersed, whereas those on other bedrocks tend to be clustered. There is no significant difference in the NTI of plant communities across different bedrocks, with NTI values being less than 0 for communities on all five bedrock types, indicating that the shallow phylogenetic structure of these plant communities tends to be clustered overall (Figure 3c). The PDI of plant communities does not show significant differences across different bedrocks (Figure 3d). However, plant communities on pyroclastic rock have a relatively high overall PDI, indicating a more complex species composition, while plant communities on granite have a relatively low overall PDI, indicating a simpler species composition.

### 3.2. Effects of Environmental Factors on Species Richness and Phylogenetic Structure

In the SAR model, climate factors and bedrock are the primary determinants of species richness in plant communities, collectively explaining 68.60% of the variation in species richness (AIC = 44.35). Isothermality (Isoth) is significantly negatively correlated with species richness, while pyroclastic rock, quartz sandstone, mean diurnal range (MDR), precipitation of driest month (PDM) and precipitation seasonality (PS) are significantly positively correlated with species richness (Figure 4a; Extended Data Appendix A). The NRI of plant communities is mainly determined by climate factors and aspects, together explaining 39.20% of the variation in the NRI (AIC = 181.12). PS is significantly negatively correlated with the NRI, whereas the aspect, temperature annual range (TAR) and precipitation of wettest month (PWM) are significantly positively correlated with the NRI (Figure 4b; Extended Data Appendix A). The primary predictors of the NTI are bedrock and four climate variables: Isoth, TAR, PWM and PDM, collectively explaining 48.60% of the variation in the NTI (AIC = 202.16). Pyroclastic lava is significantly negatively correlated with the NTI, while TAR, PWM and PDM are significantly positively correlated with the NTI (Figure 4c; Extended Data Appendix A). The primary predictors of the PDI are four climate variables: Isoth, TAR, PWM and PDM, which represent the availability of energy and water in the environment. These factors collectively explain 37.60% of the variation in the PDI (AIC = 201.18), with TAR, PWM and PDM all significantly negatively correlated with the PDI.

## 4. Discussion

### 4.1. Spatial Distribution Pattern of Diversity of Plant Communities in Qiniangshan Mountain

Our study reveals that the species richness along the elevational gradient of plant communities in Qiniangshan Mountain exhibits a “monotonically increasing” pattern (Figure 2b). Plant communities with higher species richness are primarily distributed at elevations of 400–600 m a.s.l. (Figure 2a). The reasons for the “monotonically increasing” SR are multifaceted, such as the availability of energy and water in the environment [62], human pressure in low-elevation areas [63,64] and sample strategy interference [65]. The communities surveyed in this study are well-preserved natural vegetation, largely avoiding areas of strong human disturbance. Therefore, based on the correlation between species richness and environmental factors (e.g., temperature, precipitation, bedrock types) observed in this study, we suspect that the vertical gradient of the SR of plant communities in Qiniangshan Mountain is driven by regional environmental conditions. Specifically, the high temperature and large forest evaporation in low-elevation areas lead to water deficiency, while mid-to-high-elevation areas offer the most suitable habitat due to the optimal combination of hydrothermal conditions. Therefore, the richest communities’ species composition is found there [66,67]. Furthermore, the mid-to-high-elevation areas of Qiniangshan Mountain feature more exposed rocks, which are beneficial for plants to capture moisture through rock fissures.

The overall species richness of montane plant communities on the northwest slope (inland leeward slope) of Qiniangshan Mountain is higher than that of coastal plant communities on the southeast slope (seaward windward slope) (Figure 2d). Coastal plant communities are subject to more stringent habitat filtering and ecological constraints compared to montane plant communities, such as the influence of sea winds on the windward slope, salt stress in the air and soil and nutrient-poor soil conditions [68]. The cooling effect of the ocean on rising temperatures in coastal areas may force hardwood and pine mixed forests to retreat further to the ridge tops [69]. Consequently, the species richness of coastal plant communities responds more significantly to the elevation gradient than that of montane plant communities (Figure 2d). Additionally, coastal plants are better adapted to environmental stresses (e.g., high salinity, drought and strong winds). This high specialization limits their migration to different types of ecosystems, resulting in many plant communities being confined to coastal zones, such as mangrove communities and *Glehnia littoralis* communities, which have high species endemism and conservation value [70,71].

For plant communities on the same kind of bedrock, there is no significant linear relationship between species richness and elevation (Figure 2c). This may be due to the heterogeneity of habitats created by different bedrock types, which significantly influences the distribution patterns of species richness [17]. Plant communities on different bedrock types show significant differences in species richness, with the highest overall species richness found on pyroclastic lava (Figure 3a). Pyroclastic lava is distributed in a ring at elevations of 100–600 m a.s.l. in Qiniangshan Mountain, and the soils developed on these rocks are rich in organic matter, nitrogen and phosphorus [72], providing a foundation for plant growth and development within the communities. Phylogenetic niche conservatism (PNC) posits that environmental filtering promotes the clustering of phylogenetic structures in mountainous plant communities [73,74]. Conversely, competitive exclusion drives the divergence of phylogenetic structures in mountain plant communities. The NRI results for plant communities on different bedrock types show that the deep phylogenetic structure of plant communities on pyroclastic rock tends to be overdispersed, indicating that competitive exclusion influences species’ coexistence processes within the community. In contrast, plant communities on pyroclastic lava, rhyolite, granite and quartz sandstone tend to be clustered, with habitat filtering dominating the community assembly process (Figure 3b). The NTI of plant communities does not show significant differences across different bedrock types, with the shallow phylogenetic structure of plant communities on all five bedrock types generally tending to be clustered (Figure 3c). This shows that the diverse habitats on different bedrock types play a role in environmental filtering, with habitat filtering being dominant in the community assembly of plant communities. Plant communities on pyroclastic rock have a higher PDI, indicating more complex species composition within the community, while plant communities on granite have a lower PDI, indicating smaller phylogenetic differences and closer relatedness among species (Figure 3d).

### 4.2. Climate and Bedrock Collectively Influence the Diversity Pattern of Plant Communities

One of the most urgent challenges in ecology and conservation biology is investigating how various environmental stresses affect community ecosystems and their biodiversity [75,76,77]. In this study, we revealed climate is the main factor affecting the species richness and phylogenetic structure of plant communities, and the interaction effects between climate and bedrock on species richness and phylogenetic structure should be further considered. Previous research has demonstrated a close interdependence between plant species richness and climatic variables, notably temperature and precipitation [4,78,79]. Precipitation is a key determinant of water availability for vegetation and its associated groups, which, in turn, influences plant diversity and primary productivity [5,15,80]. Biophysical processes of vegetation have been proven to be related to water availability and affect ecosystem processes [81,82]. Our findings corroborate this perspective, specifically that the climate factors MDR, PDM and PS are significantly positively correlated with species richness (Figure 4a). In addition to climatic factors, bedrock characteristics are significant predictors of species richness. Bedrock geochemistry influences vegetation growth by regulating the regolith water-holding capacity [24]. For example, volcanic rocks, due to their high porosity, have high permeability, allowing groundwater systems to enhance regional cooling effects through the bedrock layer [83]. Variations in bedrock types correspond to differences in parent soil nutrients and the capacity of rocks to retain water, together with the mountains’ microclimatic conditions. These factors contribute to habitat heterogeneity within plant communities [84], thereby providing a spectrum of ecological niches for the species that compose these communities [85]. Additionally, the diversity of bedrock types offers a range of soil substrates and micro-geomorphological environments, acting as environmental filters that influence the species composition within these communities [86].

In the SAR model, the aspect and the climate factors, TAR and PWM, are significantly positively correlated with the NRI of plant communities (Figure 4b). Aspect, as a major topographic factor in mountainous areas, affects the angle between the ground and wind direction and the amount of solar radiation received by the ground, leading to differences in light, heat, moisture and soil properties between different aspects, thereby influencing vegetation distribution [87]. This correlation indicates that aspect and climate conditions related to temperature and precipitation form an “environmental filter”, leading to the clustering of deeper phylogenetic structures within plant communities, consistent with phylogenetic niche conservatism (PNC). The climate factors TAR, PWM and PDM are significantly positively correlated with the NTI of plant communities. TAR represents the availability of energy in the environment, while PWM and PDM indicate the availability of water under extreme conditions (the driest and wettest months). The availability of energy and water determines plant diversity and primary productivity [5,15,80]. Therefore, the differences in available energy and water promote the clustering of shallow phylogenetic structures within plant communities. Pyroclastic lava is significantly negatively correlated with the NTI of plant communities, possibly because bedrocks promote the evolution of endemic edaphic specialists [88]. Habitat filtering and local species differentiation result in more distant phylogenetic relationships within plant communities on pyroclastic lava, leading to divergent phylogenetic structures. Similarly, the climate factors TAR, PWM and PDM are significantly negatively correlated with the PDI of plant communities, as these climate factors promote the clustering of phylogenetic structures within plant communities.

## 5. Conclusions

This study of the distribution patterns and environmental determinants of plant community diversity in Qiniangshan Mountain has revealed the complex interactions between local climate and bedrock in shaping species and phylogenetic structures. The peak of plant community diversity occurs at mid-to-high elevations on the mountain. Bedrock types and climate variables, particularly those related to temperature and precipitation, are significant predictors of plant community diversity. However, the importance of climate and bedrock differs for species richness and the phylogenetic structure of plant communities. It is essential to emphasize that in regions with high geological diversity, such as volcanic areas, observing patterns of species richness and phylogenetic structure requires a comprehensive consideration of climate conditions, bedrock composition and their interactions. The complex coupling effects between climatic conditions and geological substrates on plant diversity warrant continuous investigation.

## Figures and Tables

**Figure 1 plants-13-03567-f001:**
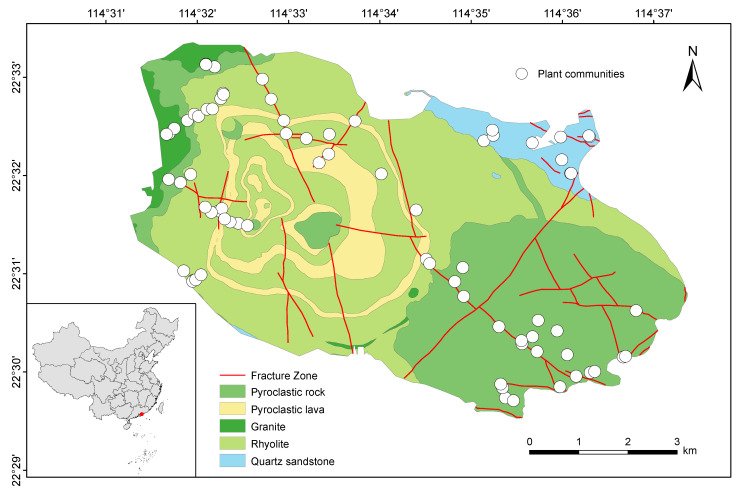
Geographical distribution, bedrock types and analyzed plant communities in Qiniangshan Mountain.

**Figure 2 plants-13-03567-f002:**
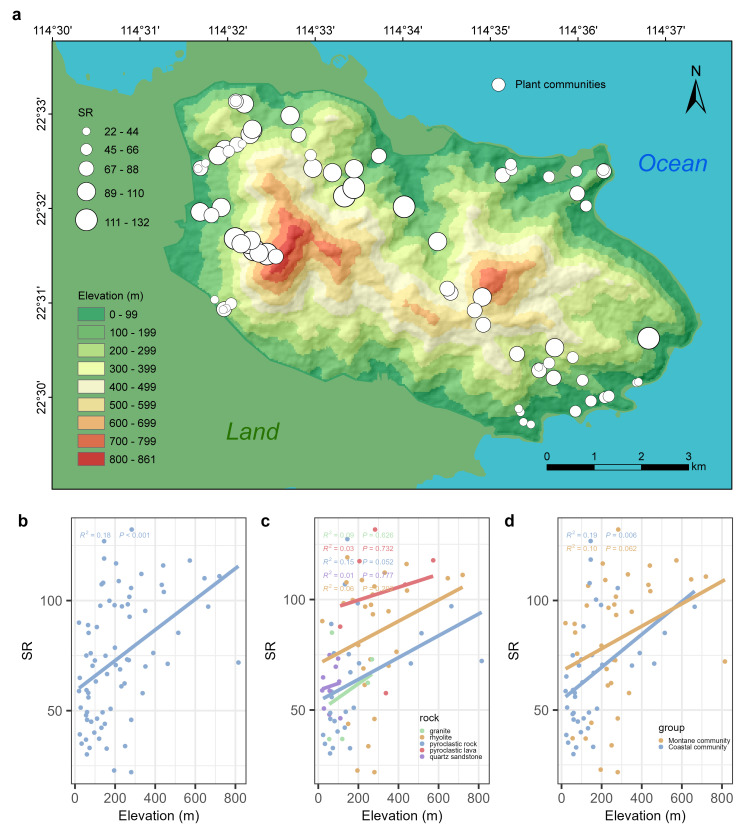
Spatial distribution pattern of species richness of plant communities in Qiniangshan Mountain and its response to elevation gradient. (**a**) Spatial distribution pattern of species richness, the size of the circle represents the SR of each plant community; (**b**) Response of species richness to elevation gradient; (**c**) Response of species richness on different bedrock to elevation gradient; (**d**) Response of species richness of montane communities and coastal communities to elevation gradient.

**Figure 3 plants-13-03567-f003:**
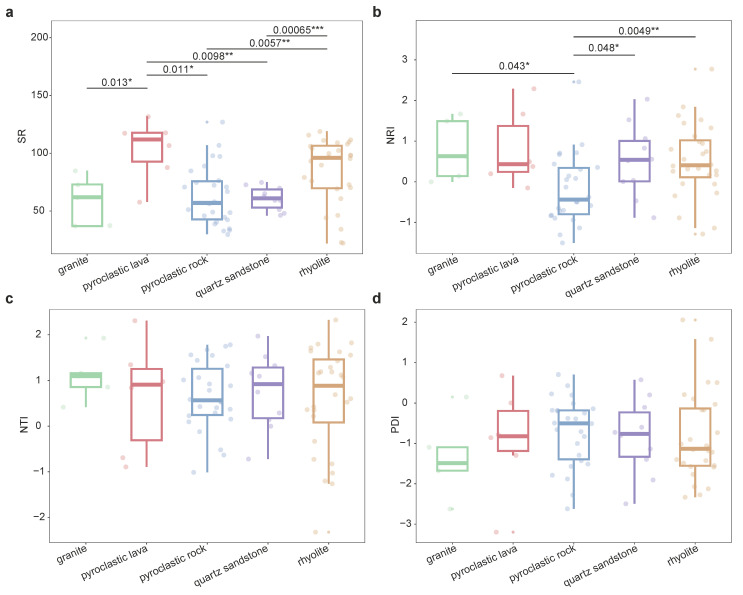
The differences in species richness, phylogenetic structure among types of bedrock. (**a**) Species richness; (**b**) NRI (Net Relatedness Index); (**c**) NTI (Nearest Taxon Index); (**d**) PDI (Phylogenetic Diversity Index); Granite, rhyolite, pyroclastic rock, pyroclastic lava are volcanic bedrocks, and quartz sandstone is a non-volcanic bedrock; T-test was used to compare the component differences. There is a significant difference between the two sets of data marked with *, * means *p* < 0.05, ** means *p* < 0.01, *** means *p* < 0.001.

**Figure 4 plants-13-03567-f004:**
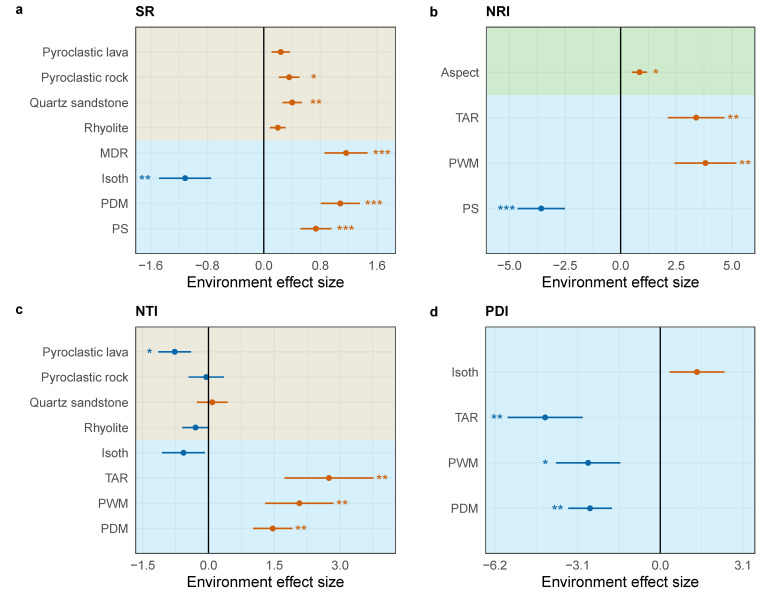
Determinants of species richness, phylogenetic structure of plant communities in Qiniangshan Mountain. (**a**) Species richness; (**b**) NRI (Net Relatedness Index); (**c**) NTI (Nearest Taxon Index); (**d**) PDI (Phylogenetic Diversity Index); Red stands for positive effect and blue stands for negative effect; Different background colors represent different environmental factors: khaki represents bedrock, blue represents climate and green represents topography; * *p* < 0.05, ** *p* < 0.01, *** *p* < 0.001. The effect sizes are calculated by SAR model. The model output results of the bedrock are calculated with reference to granite (intercept).

## Data Availability

The species list, distributions and model data of plant communities for this study can be found in the Appendix A of this study.

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
