# Peer review of "Climate and Bedrock Collectively Influence the Diversity Pattern of Plant Communities in Qiniangshan Mountain"

_plants, 2024, doi:10.3390/plants13243567_

Round 1

Reviewer 1 Report

Comments and Suggestions for Authors

The work by Xujie Li et al. aims to o explore the influence of environmental factors on the species diversity of local volcanic ecosystem in Qiniangshan Mountain, a geological remnant of the Yanshan orogeny, located in Shenzhen, Guangdong Province, China. To do so, they explore the pattern of plant species diversity and examine the importance of climatic variables and bedrock types as drivers of the taxonomic and phylogenetic diversity.

The manuscript is well written and provides a detailed description of the theoretical background underlying the study.  The methods section is well written, with only some sentences requiring a revision (please see minor comments below). I found the use of SAR models to be compelling, providing a clearer understanding of the importance of climatic and geological drivers and associations.  I have only a few major comments

1) I would suggest the authors include either a description of the climatic pattern in the study area, particularly regarding the coastal and montane slopes. While the climatic database used is referenced and described, readers will have a hard time knowing what the prevailing climate pattern in the mountain is.

2) The mountain's geological structure and elevation clearly define and constrain the distribution and availability of various bed rock types. In this regard, the SAR models seem to provide a clearer account of the observed patterns and relationships. However, I found intriguing that elevation is not included in the GLM and SAR models reported in Table S3 and Figures 2 and 3. I wonder if a GLM/SAR model that considered both elevation and montane/coastal community type in addition to the variables considered in Table S3 would allow a clearer understanding of the relative importance of these variables.  I understand this may mean a different reanalysis and major change of the manuscript's structure, and while the authors may choose to omit this avenue of enquiry, I would encourage them to discuss the potential interaction of elevation and coastal orientation or influence with climate and bedrock type. 

Minor comments

I suggest you label Figures 3 and 4 so that readers may easily know whether the model being examined is a GLM or SAR model. This is particularly relevant for the results in Figure 4, which seem to be related to the results in Table S3. However, SAR models are not always the most parsimonious. (See models for NRI in Table S3). 

Please revise and edit the phrase in lines 203-204, which seems to end abruptly. A possible alternative is to merge with the next sentence in lines 204-206.

I suggest you provide a reference for the Hellinger transform at the end of sentence in lines 223-224. A possible reference could be Legendre, P., & Gallagher, E. D. (2001). Ecologically meaningful transformations for ordination of species data. Oecologia, 129, 271-280.

 Please revise the sentences in lines 228-229, which seem to make more sense if merged in a single sentence and also seem redundant with what you state in lines 230-232. 

Line 296: I suggest you edit "quartz sandstone is non-volcanic bedrocks" to "quartz sandstone is a non-volcanic bedrock"

Author Response

“Please see the attachment.”

Reviewer 2 Report

Comments and Suggestions for Authors

This study used the Qiniangshan Mountainas a model to investigate the complex interactions between local climate and bedrock in shaping species and phylogenetic structure. They recorded 75 plant communities, and showed that the peak of plant community diversity occurs at mid to high elevations on the mountain. The variables related to temperature and precipitation are significant predictors of plant community diversity. In addition, the importance of climate and bedrock differs for species richness and phylogenetic structure of plant communities. Their results are significant to explore the species richness under high geological diversity.

However, the figures through text are not clear enough.

Author Response

Thank you very much for taking the time to review this manuscript. When we submitted the manuscript, we provided clear enough original pictures, and the resolution of each picture was above 300 dpi. 

Reviewer 3 Report

Comments and Suggestions for Authors

The manuscript deals with the influence of climate and bedrock on plant communities diversity and phylogenetic diversity in a Chinese site. The topic is interesting. The results needs to be implemented with a more precise description of the obtained results on the phylogenetic diversity. According to the guidelines for authors of the journal, in the text the literature should be cited using progressive numeration in square parenthesis, and the references list should not follow an alphabetic order, but the numeration in the text. I suggested some minor corrections and improvement directly on the file in the attachment.
